# Plant Growth Regulation in Cell and Tissue Culture In Vitro

**DOI:** 10.3390/plants13020327

**Published:** 2024-01-22

**Authors:** Taras P. Pasternak, Douglas Steinmacher

**Affiliations:** 1Instituto de Bioingeniería, Universidad Miguel Hernández, 03202 Elche, Spain; 2AlfaPalm Agrociências, Marechal Cândido Rondon 85960-148, Brazil

**Keywords:** auxin, biotechnology, culture media, somatic embryogenesis

## Abstract

Precise knowledge of all aspects controlling plant tissue culture and in vitro plant regeneration is crucial for plant biotechnologists and their correlated industry, as there is increasing demand for this scientific knowledge, resulting in more productive and resilient plants in the field. However, the development and application of cell and tissue culture techniques are usually based on empirical studies, although some data-driven models are available. Overall, the success of plant tissue culture is dependent on several factors such as available nutrients, endogenous auxin synthesis, organic compounds, and environment conditions. In this review, the most important aspects are described one by one, with some practical recommendations based on basic research in plant physiology and sharing our practical experience from over 20 years of research in this field. The main aim is to help new plant biotechnologists and increase the impact of the plant tissue culture industry worldwide.

## 1. Introduction

Plant tissue culture and plant regeneration constitute crucial facets of plant biotechnology, spanning several scientific and industrial domains. In recent years, we have witnessed significant advancements in the comprehension of the underlying mechanisms controlling in vitro plant regeneration, accompanied by the rapid evolution of specialized equipment and strategies for enhancing this process. This development has garnered the attention of both plant scientists and industries. Nonetheless, there exists an ongoing demand to further develop our strategies to select and regenerate superior genotypes, thereby refining protocols for enhanced reproducibility. In this context, a meticulous fine-tuning process is imperative. Plant tissue culture has several regenerative pathways. The development and application of cell and tissue culture techniques are usually based upon empirical studies. Overall, the success of PTC (plant tissue culture) is dependent on several factors, including optimal nutrient balance and the presence of competent cells, which can perform certain epigenetic and molecular re-programming for stem cell induction and precise spatial and temporal regulation of endogenous hormone synthesis and its distribution.

New approaches using data-driven optimization of culture medium composition also show very relevant results [1], but it is shown in this communication that the directed use of current knowledge on classical plant physiology can result in comparatively rapid elucidation and marked improvement of culture techniques. Moreover, many of the presented results and observations are also based upon the practical experience of the authors, gained during more than 30 years of research in this field. The main objective of this review is to show that plant biotechnologists and related industries can use this manuscript to find a way to circumvent unnecessary work on symptoms by focusing on the fundamental regulatory mechanisms of plant development. In the present review, we are devoted to descriptions of all these steps one by one with some practical recommendations.

## 2. Major Aspects of Plant Regeneration

### 2.1. Regenerative Pathways

#### 2.1.1. Organizer Cell Type Controlling Plant Regeneration

By being sessile, plants have evolved with a complex network of structures and signals to regulate and coordinate their development. One of these strategies is the occurrence of stem cell niches which might develop into a new organ or even a new plant, upon certain conditions. Nowadays, plant shoot (less) and root meristems are by far the most studied stem cell niches, but plant tissue culture has basically evolved from this observation and the occurrence of this phenomenon [2,3]. However, in this review, we postulate the de novo formation of the organizer cell type also coordinating plant cell fate with in vitro responses on shoot pathways. This is compatible with the concept presented by Mayer et al., (1998) [4], which suggests that although there is ample evidence that shoot and root meristems employ different sets of regulatory genes, both stem cells may be specified by a central organizer cell. In roots, the organizer cell corresponds to the quiescent center, and WUSCHEL and its HOMEOBOX (WUS–WOX) expression is a marker for this cell type [5,6]. Also, in the root model and using different strategies, Sabatini et al. [7] have shown that cells from different types of tissue are competent to form organizer cell types in response to high levels of auxin. Evidence has proven the interconnection between auxin and WOX5 pathways, and recent studies have also shown that WOX5 induces TAA1-mediated auxin biosynthesis [8]. Therefore, auxin biosynthesis and accumulation are crucial facets for plant regeneration. In the root, the stem cell niche is formed as a result of auxin accumulation in the xylem pole pericycle cells, through xylem-mediated (as the initial stage) and active auxin transport responsible for stem cell formation [9]. In the shoot stem cell, the opposite situation occurs. A group of small cells formed either from callus or from an organ’s surface (direct regeneration), this group can be named as the organizer cell niche and should produce its own auxin from an auxin gradient. These cells (WUS active) are relatively dormant and have a special nuclei feature (compact nuclei with small nucleoli, histone hyperacetylation) [10]. Organizer cells can also be formed from the competent cells in plant organs, which do not connect with the xylem/vasculature and cannot canalize auxin in this way. In the case of xylem formation, only the root can form. Showing that this set of genes is essential for plant regeneration, ectopic WUS expression might, therefore, be a relevant strategy for enhanced plant regeneration. For further details see [11].

#### 2.1.2. Organogenesis

Organogenesis is the most commonly applied micropropagation technique for a massive (large-scale) plant propagation method, using relatively small plant tissue explants from the mother plant [12]. These plants grow in vitro in sterile conditions under applications of growth regulators. Micropropagation can be scientifically divided into three steps, each of which requires significantly different conditions: 1. de novo induction of “organizer cell”, which is required for de novo shoot formation; 2. growth of new shoots by creating auxin gradients followed by cell specification, induction of multiple cell fate-specific auxin biosynthesis pathways with specific chromatin organization in each cell type; 3. induction of root from formed shoot by creation of temporary auxin maximum in xylem-adjusted cell, cell cycle activation and directional auxin transport. Growing mother plants on optimized culture medium led to an increase in tissue response and a rise in the propagation ratio.

#### 2.1.3. Somatic Embryogenesis

Somatic embryogenesis (SE) is the induction de novo of embryos from somatic plant cells. This morphogenetic pathway can be induced by a number of auxinic herbicides, 2,4-Dichlorophenoxyacetic acid (2,4-D), Picloram, and Dicamba, at a relatively high concentration [13]. These synthetic auxins, in addition to their auxinic effect, bind to F-box proteins as transport inhibitor response 1 (TIR1) [14] and can induce endogenous auxin biosynthesis, but can slightly inhibit its efflux. The competent cells, presenting regular ploidy and regular nuclei structure, and are able to start auxin biosynthesis (via TAA1-IPA pathway) performing initial auxin accumulation, results in the formation of “organizer cell”, which, in the future, divide periphally to form globular structures with surrounding epidermis [15]. However, such a system is not very effective since non-embryo competent cells also enter in cell division and form callus because of exogenous auxin. In this case, the resources available from the culture medium can dissipate as the division rate of callus cells is usually higher. One simple way to improve this step might be to slightly reduce auxinic herbicides’ concentration and increase auxin endogenous accumulation in the organizer cells with the inhibition of auxin efflux (pH, salicylic acid (SA), Iron (Fe), Copper (Cu) at a certain concentration). In this case, the effect of 2,4-D and other synthetic auxins will be more targeted, avoiding callus formation and accelerating organizer cell formation, thus increasing their number and, therefore, the plant regeneration capacity.

#### 2.1.4. Haploid Induction In Vitro

Using the doubled haploid system is the simplest and fastest method of homozygotic hybrid production, a very important strategy for breeding programs. The most used method is microspore embryogenesis. This method has been established for many important crops (barley, pepper, rapeseed, rice, sugar beet, and wheat). The physiological mechanism of microspore embryogenesis is similar to that of classical somatic embryogenesis and requires stress factors and auxin interactions with further epigenetic modifications to switch from gametophytic to sporophyte pathways [16].

#### 2.1.5. Protoplast Culture

Plant protoplast culture has two main applications: somatic hybridization, including asymmetric somatic hybridization [17], and the study of cell totipotence [18]. Somatic hybridization is especially important for hybridization between species or genera that cannot be forced to cross-breed using traditional sexual hybridization techniques. In addition, protoplast can serve as a powerful tool for testing different gene constructions in transient assays, including modern techniques of target gene editing like CRISPR. In this case, plant regeneration is not required [19,20].

However, so far, for plant regeneration, protoplast cultures are rarely used since, contrary to plant regeneration from explants, they require much more effort and are strongly genotype dependent [18].

### 2.2. Nutrient Balance In Vitro

Although we are facing the advent of new approaches and knowledge of molecular aspects of plant regeneration, mineral nutrients form a significant component of culture media but are often overlooked as possible morphogenic elicitors [21]. It is well known that each ion is a possible morphogenic factor/elicitor [22]. Recent investigations indicate that the signaling pathways for the majority of nutrients are subject to epigenetic regulation [23,24,25]. Therefore, optimal nutrition is key to the success of plant tissue culture, and when choosing the culture medium, one needs to consider the role of each element/ion in morphogenesis based on classical plant physiology [26].

Interestingly, some popular culture media based on another nutritional concepts have been adjusted for soft callus and, after that, mistakenly extrapolated to whole plants. The most famous culture medium used in plant micropropagation is Murashige and Skoog medium (1962) (MS) [27], and it presents the following formula: N-K-Cl-Ca-S-Mg-P-Fe-Mn-B-Zn-Mo-I-Cu-Co with an N:P ratio of 40:1. It is important to mention that the original purpose of the this formula was to develop a revised medium for the rapid growth of tobacco cells from pith tissues and was not focused on plant regeneration and plant quantity. This formula made halides (chloride) a major anion after nitrate, which in turn, based on the ability of chloride-induced rapid post-mitotic cell expansion, resulted in rapid soft callus growth. Nowadays, several authors, who also work on the optimization of the composition of culture media, have also suggested that the MS formula is far from optimal for micropropagation (plant regeneration) and rooting [28,29].

Additionally, all nutrients/ions in each culture medium can be divided into four main groups, namely major macros (N, K, P), minor macros (Ca, Mg, S), mediate macros (Fe) and micros (Mn, B, Mo, Zn, I, Cu, Co). Macro nutrients serve as structural components or are involved in several metabolic reactions (N, P, Ca), while microelements serve as cofactors of numerous hormone- and stress-related reactions/responses, with significant interactions occurring among those nutrients. The basic nutritional formula for plants/plant cells is as follows: N-K-P-Ca-Mg-S-Fe-Mn-B-Zn-Cl-Mo-Cu-Co-I with an N:P ratio of (5–12):1. This formula is based on results from more than 100 years of research on plant physiology and considers the role of each element. In addition, is necessary to consider a very complicated interaction between micro and macro nutrient in terms of cell uptake and cell signaling [30].

In summary, it is important to note that in the case of plants growing in the soil, ions present mainly in the form of complexes but not in pure form. Therefore, most of the plant nutrition investigation was carried out in water/hydroponic cultures. Since plant cells respond in completely similar ways in vitro and in nature, the results from the hydroponic culture were easily extrapolated to the in vitro system. Similarly, the molecular and physiological mechanisms of the stress response were very similar in vitro and in soil, which further pointed out the similarities between in vitro and in-soil systems. Below, we discuss some specific points related to each nutrient, which we consider relevant for developing efficient strategies for plant regeneration.

#### 2.2.1. Macro Elements

In natural conditions, plants require six macro elements captured from their roots (N, K, P, Ca, Mg, S), while carbon (C) is obtained from air and fixed as a product of photosynthesis. However, in vitro, plants may require carbon supplementation as well. Next, we briefly describe the function of each element and their optimal concentrations.

##### Carbon

Carbon (C) is a main component of almost all macromolecules in the plant body (cell wall, protein, DNA, RNA). Carbon starvation is lethal for plants from the onset of seedling growth [31]. Some different types of carbohydrates, including sucrose, glucose, and maltose, can be used for in vitro plant regeneration. Sucrose is the most useable compound since it is inexpensive and resistant to autoclaving.

Glucose is the most used carbon source by plants in nature; however, there are significant disadvantages to using it directly for in vitro plant regeneration. Firstly, glucose is unstable in solution at pH 5.6 and can spontaneously open the ring to convert to lactic acid. This process is significantly accelerated during autoclaving [32], and we are strongly recommended to use only fresh filter-sterilized glucose. Conversely, sucrose is relatively resistant to autoclaving but can be caramelized after prolonged heating. Sucrose concentration depends on the task: 1% can be for photosynthetic plants and up to 6% can be sued for embryogenic callus. Usually, 2% to 6% is used for initial plant growth. This concentration cannot be considered osmotic and has a metabolic function as a carbon source for in vitro plant growth [33].

##### Nitrogen

Nitrogen (N) is key to plant nutrition and its concentration in media must be the second highest after carbon. Nitrogen is a major component of amino acids (proteins) as well as other important biological macromolecules. The form of nitrogen is also extremely important for plants. On the whole plant level, the most beneficial form of nitrogen is strongly dependent on the pH. Ammonium is directly involved in metabolic pathways, while nitrate requires conversion to ammonium through nitrate and nitrite reductase activity. Thus, on one hand, ammonium is beneficial for plant nutrition. On the other hand, the application of ammonium only to plant cells leads to extremely quickly uptake and nitrogen overloading [34,35,36,37]. Several recent investigations have suggested that nitrogen signaling is directly linked with auxin signaling [38,39] and epigenetic modifications [36]. The optimal nitrogen concentration for the majority of plant species is ~15 mM N, with a nitrogen to phosphate (N:P) ratio of 5–6:1 for shoot induction and 10–12:1 for plant growth.

##### Potassium

Potassium (K) is the third most important ion that plants require. Contrary to nitrogen, potassium is not a structural element but plays an essential role in plant development. Potassium is a symporter with hydrogen and can increase auxin transport. The first link between auxin distribution and potassium was reported in 1962 [40]. Later, a detailed investigation of potassium nutrition led to the conclusion that potassium deficiency leads to a significant alteration in auxin distribution [41] through several potassium transporters. A recent review by Johnson et al. [42] further confirmed the role of potassium in the regulation of cell-to-cell communications and stress responses in plants. Additionally, potassium is used for agar-solidification and an extra source of K can reduce hyperhydricity and promote cell division, especially when it is caused by auxin deficiency (barley microspore, wheat ovary, etc.). We suggest using at least two times less potassium than nitrogen and even less for induction [43]. Potassium is an ion with extremely high mobility in plants and plays a key role in regulating cell-to-cell communication and transport (N, C), including auxin transport. In many cases, high potassium nutrition alleviates stress effects [44]. Based on these properties, potassium positively affects plant growth in vitro during de novo shoot development, rooting, and by avoiding hyperhydricity. However, high potassium levels may have a negative effect on the induction stage, while potassium deficiency or low levels can be useful for creating organizer cells, which should be semi-autonomous (personal observation).

##### Phosphate

Phosphate (P) is the fourth element in the classical plant nutrition formula and plays a role as a structural element (phospholipid, DNA) and as a metabolic molecule involved in phosphorylation/dephosphorylation, ATP/ADP transition, etc. [45,46]. In contrast to nitrogen, P is much more labile (it is only non-utilizable in membranes). There are many reports on how phosphate nutrition regulates processes in PTC [47]. However, we suggest using an optimal P concentration (2–3 mM), keeping the N:P ratio as described above for the nitrogen part.

##### Calcium

Calcium (Ca) is a known signal transducer involved in the regulation of many kinases, the cell wall structure, and responses to stimuli. Calcium plays an important role in plant tissue culture [48,49] through its involvement in different signaling processes. The normal concentration of Ca in culture media is the same as that of P and should be around 2–3 mM. In this respect, the calcium source is key to plants’ in vitro regeneration. In the majority of cases, based on MS culture medium, researchers use calcium chloride (CaCl_2_) as a Ca source and even studied the effect of removing Ca or adding extra CaCl_2_. This, in turn, made chloride (Cl) a major anion after nitrate with its numerous negative effects on PTC (for details, see chloride section). Therefore, we suggest using calcium nitrate [Ca(NO_3_)_2_] as a Ca source and balancing other media components instead, keeping Cl at a low concentration, as for the minor nutrient class.

##### Magnesium and Sulphur

Deficiency of both magnesium (Mg) and sulphur (S) components has a very negative impact on plant development in soil since both are involved in many vital reactions as cofactors (Mg) and as structural components (S) as parts of amino acid and the central hub of thiol [50]. The optimal concentration of both components is around 1.5 mM, so magnesium sulfur (MgSO_4_) is an ideal salt for PTC. It is also important to note that both Mg deficiency and Mg excess have significant negative effects on plants and plant cells, for example, on Prunus micropropagation [51].

In conclusion, for the macro elements (excluding carbon), we suggest to use the following four salts: potassium nitrate (KNO_3_), ammonium phosphate monobasic (NH_4_H_2_PO_4_), calcium nitrate (Ca(NO_3_)_2_), and magnesium sulphate (MgSO_4_). These salts perfectly fit to create optimal ion concentrations in the medium.

##### Iron

Iron (Fe) is an essential macro element for plants [52,53]. Iron plays an important role in accepting and donating electrons as well as an essential role in the electron transport chains of photosynthesis and respiration. The optimal concentration of iron in the medium is 70–100 µM, in the form of iron(II) sulfate heptahydrate (FeSO_4_·7H_2_O), a commonly used salt. However, Fe is not soluble and is rarely accessible to plants; therefore, it is used more effectively in a Fe chelate form, as ferric ethylenediaminetetraacetic acid (FeEDTA) chelated from FeSO_4_, or in a directly chelated form, as ethylenediamine di-2-hydroxyphenyl acetate ferric (FeEDDHA), among others [54]. The chelated form FeEDDHA is more stable at different pH levels and has shown relevant results for different species evaluated, including date palm [55], walnut [56], and hazelnut [57]. It is also important to note that excess iron may cause iron stress, which can effect embryogenesis [58]. As expected, the iron effect directly relates to epigenetic regulation [59]. However, one must consider that iron is one of the elements that easily precipitates in a culture medium. Below, in the section “Plant medium preparation”, we describe this point in more detail.

#### 2.2.2. Microelements

##### Boron

The primary functions of boron (B) relate to cell-to-cell communication and its structural role in the cell wall [60]. Its roles include involvement in membrane integrity, seed production, root elongation, and sugar metabolism [61]. Pioneering work on the boron deficiency effect was performed in 1920 [62,63]. A recent investigation pointed out the role of boron at the molecular level also during fruit development [64] and its link with auxin transport/metabolism. Boron is not a re-utilizable element, and it accumulates in the rigid cell wall of mature cells. In PTC, low boron levels may be useful for the de novo shoot induction stage, as a critical minimum content in the meristematic cells is necessary to elicit mitosis [65]. However, given its essential role in different aspects of cell maintenance and plant growth, including interaction with hormonal pathways [66], reaching this minimum threshold is not an easy task, and considering that the earliest defects following B starvation are growth arrest and aberrant meristem formation [65], it is better to keep the boron concentration constant during all stages of PTC.

##### Manganese

Manganese (Mn) is a metal and an essential cofactor for the oxygen-evolving complex (OEC), playing a role in photosynthesis and many other functions [67,68]. While Mn deficiency/excess affects plant morphogenesis, the difference between its thresholds of deficiency and toxicity is very small, as shown in Nicotiana [69]. Therefore, if the objective is not to test the manganese effect, we do not recommend interfering with it because the effect can be very complicated. A higher Mn concentration can help in the induction stage, but generally, excess Mn is highly toxic [70]. The recommended concentration of manganese in the medium is from 20 to 100 µM, and researchers should consider using manganese chloride (MnCl_2_) as a salt source, keeping both ions at a micronutrient concentration.

##### Molibdenium

Molibdenium (Mo) is a trace element required for plants in micro doses [71]. Mo-containing enzymes play a role in the redox cycles of nitrogen, carbon, and sulfur. Mo remains biologically inactive until it becomes complex enough to form a Mo-cofactor [72]. Based on the classical culture medium composition, Mo concentration is around 0.2 to 1 µM. So far, there are no published studies on the effect of excess or deficient molybdenum. So, we suggest keeping the concentration in the range of 0.5 to 1 µM.

##### Copper

In plants, copper (Cu) is a cofactor of many reactions and was first described in 1939 [67,73]. Also, Cu serves as an epigenetic regulator [74]. The optimal concentration of Cu is around 1–2 µM (not 0.1 µM as in the medium for callus). Increasing the copper concentration to 20 mg/L (80 µM) had a positive result for plant regeneration in several plant species [75]. Cu stress (20–30 µM) may serve as a trigger of cell division and plant regeneration [76].

##### Zinc

Zinc (Zn) is an essential ion that serves as a cofactor of Cu/Zn Superoxidedismutase (SOD) in chloroplasts and cytoplasm. In addition, investigations found the important role of Zn as a catalysator of auxin biosynthesis [77,78,79]. The optimal concentration of Zn is around 5–10 µM as ZnSO_4_ salt.

#### 2.2.3. Halogens

The halogens group is, in nature, represented by three major members: chloride (Cl), bromide (Br), and iodine (I). So far, two (Cl and I) have been proven as micronutrients essential for plant development.

##### Iodine

Iodine (I) was recognized as a micronutrient as early as 1961 [80]. Recent investigation showed that iodine modulated gene expression, interacted with protein by iodination, activated multiple metabolic pathways, and was mostly involved in defense responses [81]. For a review, see [82]. The optimal concentration of iodine in plant culture medium is in the range of 0.2–5 µM. While variation in iodine concentration may have some effects on plant tissue culture for specific species, such as Cocos nucifera [83], the effects may be very complicated. We suggest keeping the concentration around 1–2 µM.

##### Chloride

Chloride (Cl) is the most contradictory ion in plant tissue culture in vitro. Chloride is a halogen element recognized as a micro nutrient and plays a specific role in proton transfer reactions in photosynthesis [84]. However, chloride can accumulate in other acidic compartments besides chloroplasts which further increases its acidity. This function is explored in plants in two cases: in micro doses (10 µM), it can be useful for acidification of the stroma to enhance the rate of photosynthesis. This function is not very relevant for in vitro culture regarding carbohydrate supply. In macro doses (6 mM in MS culture medium [27], up to 34 mM in Y3 culture medium [84]), chloride is rapidly taken up in the middle acid vacuole (gamma-type),and helps to convert it to a lytic vacuole with increased turgor pressure. This application is relevant if soft calluses are desired, as in [27,85] or it might have a role in plant cell elongation [86]. However, although an effect might be seen on the plant growth rate, this might be linked to higher water uptake by plantlets [87], and due to in vitro conditions, it might also result in hyperhydric plantlets with a lower ex vitro survival rate. Therefore, for in vitro plant regeneration, chloride should be omitted or used only as a micronutrient with 20–100 µM concentrations during the plant/root elongation phases.

### 2.3. Organic Components

#### 2.3.1. Bacto-Tryptone

Bacto-tryptone (casein hydrolysate) is an extremely important culture medium component since it prevents ion precipitation [88]. In our case, we have used this compound not as a nitrogen source and not as an amino acid mixture, but only as a compound to prevent precipitation. We suggest using 100 mg/L (which is usually equal to a max 13 mg of nitrogen) and a very low concentration of some amino acids, which can be ignored as a nutrient source.

#### 2.3.2. Ascorbic Acid

Ascorbate or ascorbic acid (ASC) is a low-molecular-weight antioxidant and can be considered a “paradoxical” compound [89]. In plants, the main ascorbic acid pool is related to mature organs, linked with plastids (APX), and has a low level in the meristematic zone [90,91]. Ascorbate biosynthesis-deficient mutants do not demonstrate visible phenotypes during morphogenesis in plants. Changes in the endogenous ASC pool, as well as the effect of exogenous ASC on plant cells in vitro, clearly demonstrated that ASC is a negative regulator of cell proliferation but can protect mature cells from damage and from excess phenolic compound accumulation during callus growth [92]. Adding ASC directly to culture medium has some direct implications. At pH 5.6, ASC’s half-life is just a few hours, after which it converts to dehydroascorbate (DHA) and further to oxalic acid [93]. Moreover, ASC is a highly charged molecule that plant cells cannot directly absorb. So, first, ascorbate is oxidized to DHA, it is absorbed by plants in this form, and as a neutral molecule, it is rapidly absorbed and immediately converted back to ASC by DHA-reductase within plant cells. Once ASC reaches the cytoplasm, it becomes stable since cytoplasmic pH is around 7. To avoid the side effects of ascorbic acid product degradation, we suggest using short-term (up to 6–10 h) incubation of explants/callus in DHA solution (300–500 µM) and, after that, transferring plants /callus to a new medium without DHA.

#### 2.3.3. Glutathione

Glutathione (GSH) represents a class of thiol and is considered an antioxidant. However, despite this function, GSH also regulates auxin response [91,94,95,96]. In plants, GSH exists in several isoforms. Such divergence is related to possible different GSH functions. In the Fabaceae species, GSH significantly promotes cell cycle progressions and plant regeneration [94,95]. GSH application in PTC requires special attention since at millimolar concentrations, it serves as a strong buffer in a pH range of 4–4.5 [90] and requires adjustment in the pH level after addition.

#### 2.3.4. Amino Acids

Several authors suggest adding all 20 amino acids at low concentrations to the medium. However, there are better decisions than this. Among amino acids, we suggest using glutamine, which, in turn, serves as an epigenetic regulator (see below) at a concentration of 20 mg/L or similar. This concentration is widely used for the improvement of embryogenesis in several plant species [96,97]. The auxin precursor tryptophan may also positively affect some plant species [98].

#### 2.3.5. Sodium (Potassium) Humate

Humic acid has a very positive effect on plant productivity in soil by alleviating many kinds of stresses; see [99] for a detailed description. In tissue culture, neutralized sodium (potassium) salt of humic acid has been used at a concentration 5–15 mg/L mainly for the improvement of explant rooting. This protocol is very effective for root formation and growth in some recalcitrant species and is highly recommended [100,101,102].

#### 2.3.6. Vitamins

Vitamins are necessary compounds synthesized and utilized in plants. The effects of vitamins on plants were first described more than 80 years ago [103,104]. In tissue culture media, the effects of different vitamins were first studied by Linsmaer and Skoog (1965) [105]. The addition of exogenous vitamins has a significant positive effect on plants in vitro (for review, see [106]). Among vitamins, B1 and B6 are two of the most important ones and are widely used.

#### 2.3.7. B1 as Auxin Cofactor

B1 is the most useful vitamin since it serves as a cofactor in endogenous auxin biosynthesis [104,105,106]. Linsmaer and Skoog (1965) [105] demonstrated the exclusive role of thiamine (B1) in tobacco callus growth. Digby, J. and Skoog, F. (1966) demonstrated the induction of B1 synthesis in plants by cytokinin as an auxin biosynthesis inducer [107]. Gamborg (1968) proposed an increase in B1 content to 10 mg/L, and currently, this concentration is widely used in PTC [108]. An even higher concentration of B1 (however, not in PTC) showed superior results in Jatropha clonal propagation [109].

#### 2.3.8. B6 as a Promoter of Rooting

B6 plays a key role in auxin homeostasis in roots [110] and can be helpful for de novo rooting. An amount of 1–2 mg/L of B6 is a standard concentration in the medium. Other vitamins may play a specific role, and in some cases, “vitamin stress” may positively affect PTC. Nevertheless, playing with these compounds may be quite complicated, and we suggest using the basic vitamin formula B1:B6:PP (nicotinic acid) in a 10:1:1 ratio (in mg/L) [111].

### 2.4. Phytohormones

Exogenous phytohormones serve as a tool for the regulation of different processes in PTC. That is why most researchers mention exogenous phytohormones as regulators of morphogenesis. However, plant morphogenesis is regulated exclusively by endogenous hormones, and exogenous ones can serve as modulators of their action only.

#### 2.4.1. Auxin(s)

Auxin is an essential hormone responsible for all processes in PTC. It is the only exclusive hormone that can transport polarly, create gradients, and regulate cell fate. So, auxin is responsible for both shoot and root morphogenesis. In plants, local auxin biosynthesis was responsible for one or more morphogenic processes [112,113,114]. For a review, see Verma S. et al., 2021 [115]. The next question is about endogenous or exogenous auxin. Recently, it was shown that the main role in PTC belongs to endogenous auxin [116,117]. Since plants are represented as having continuous auxin gradients with different origins, applying exogenous auxin in PTC leads to disturbance in plant morphogenesis and should only be carried out in the induction stage. Among exogenous auxins (auxin analogs), 1-Naphthaleneacetic acid (NAA) is the closest to natural Indole-3-acetic acid (IAA) in terms of its function but is more stable; other analogs like 2,4-D, 2,3,5-T, dicamba, and picloram can be considered rather as “auxinic herbicides”, can induce endogenous auxin synthesis [118], and have some stress effects. That is why “auxinic herbicides” are widely used for somatic embryo induction.

Auxin is the most common plant growth regulator used to induce somatic embryogenesis [119]. It has been shown that auxins act like molecular glue, binding to their TIR1 receptor and promoting ubiquitin-dependent degradation of Aux/IAA repressor proteins [114,120]. Naturally occurring auxin (IAA) and synthesized auxin analogs (NAA and 2,4-D) showed the same activity [14,120]. In plants, auxin biosynthesis is represented by several pathways, the main one of which is tryptophan dependent. In the frame of this pathway, first, Indole-3-pyruvic acid (IPA) was formed by the action of the TAA1 gene, and thereafter, numerous YUCCA genes were converted from IPA to IAA. Recently, the key role of the TAA1 gene and several auxin transporters in the process of the induction of embryos in plants was shown [120,121]. However, auxin itself, auxin “concentration” or level, cannot be used as a marker for any morphogenic processes. In plants, several “homological” genes (YUCCA, TAA1) are responsible for auxin production, and each plays its own role. In tomatoes, for example, there are at least seven YUCCA genes; each of them shows activity in different stages of cotyledon development and they cannot replace each other [122].

In tissue culture, exogenous 2,4-D induced endogenous auxin biosynthesis, which may relate to further auxin independence and somatic embryo induction [58]. Later, it was shown that the LEC2 gene promoted the embryogenic pathway through auxin biosynthesis and the role of TAA1 in this process was pointed out [123]. There is some evidence about the role of TAA1 in plant regeneration. First, this includes the exact colocalization of TAA1 with the shoot organizer cells [124] with potentially competent cells in the leaf. Second, TAA1 is colocalized with the wox5/wus gene [8], responsible for shoot regeneration [125]. Third, WUS (stem cell organizer) is induced by auxin [126].

In summary, de novo shoot formation requires the presence of the auxin source in a specific cell type (organizer cell, which divides very slowly in the center and has a specific chromatin status). These cells synthesized IPA through the TAA1 pathway, which further converted to IAA and influenced the action of YUCCA 2. Cells on the periphery of the cluster divide faster and possess auxin canalization through PINs action to form an auxin gradient, which, in turn, leads to epigenetic modification with the formation of different cell types. Endogenous auxin also plays a primary role in the rooting step as part of the plant micropropagation procedure. Namely, through the root is considered the simplest means of auxin canalization, and the root can form only after the formation of a sieve element vessel. In PTC, pulse application of exogenous auxin can induce de novo root primordia in the already formed xylem pole “pericycle” cell, which can be realized after removing exogenous auxin. There are two key factors for successful rooting: a high rate of auxin biosynthesis in new shoots and a high rate of auxin flux from these shoots through the vessel. Nutrient medium composition is critical in this case. A possible model of plant regeneration is summarized in Figure 1.

#### 2.4.2. Cytokinin(s)

The main effect of cytokinin in tissue culture in vitro is shoot induction [127], which occurs through auxin biosynthesis induction and maintenance [128,129]. There are several natural cytokinins [Zeatin (Zea) and its riboside (Zea-R)] and artificial cytokinin-like compounds [Benzilaminopurine (BAP), Kinetin (Kin), 2-isopentenyladenine (2-IP)]. In addition, Thidiazuron (TDZ; N-phenyl-N′-1,2,3-thiadiazol-5-ylurea) was recognized as a potent regulator in vitro as early as 1990, with cytokinin-likes responses [130]. In addition to its cytokinin-like effects, TDZ can also be considered as a stress factor [131]. Currently, TDZ is used successfully in many plant species (for a review, see [132,133]). Cytokinin action/effect is dependent on the ability of the cell to produce endogenous auxin by one pathway or another. Only the competence of the cell to activate the TAA1 pathway (linked with the “organizer” cell) can induce shoots. A direct link between TAA1 and auxin has been confirmed by the presence in the TAA1 promoter of two cis elements of ARR1 response. Therefore, transcriptional regulation of TAA1 serves as a prerequisite for further regulation of auxin (IPA) biosynthesis. In the majority of dicotyledon shoot explants (leaf/cotyledon), exogenous cytokinin induces either callus or shoots or roots. These phenomena represent three faces of cytokinin, which are dependent on the origin and type of auxin biosynthesis pathway. Namely, in tomato cotyledon, the “soft callus” forms mainly from mesophyll cells (direct organogenesis), and somatic embryogenesis forms from cell layers near the epidermis with the active TAA1 gene, while auxin canalization to the xylem leads to the formation of large sieve elements with further possibility of root induction after the exogenous cytokinin level is lowered [134]. However, in monocotyledon, leaf auxin transport is directed from the meristem to the leaf tips (opposite to dicotyledons), so the leaf is not able to produce endogenous auxin and, therefore, cannot react to exogenous cytokinin application.

#### 2.4.3. Other Hormones

##### Salicylic Acid

Salicylic acid at least partially performs specific action as an inhibitor of cell-to-cell communications. This function, indeed, is very useful in PTC since it promotes auxin accumulation in putative stem cells (during the induction phase) and should be used in low–middle concentration (25–100 µM) in combination with endogenous auxin synthesis (induced by cytokinins) or exogenous cytokinins [135,136,137,138]. However, prolonged incubation with high SA inhibited embryo development [139]. A recent investigation demonstrated a positive effect of 40 µM SA on olive somatic embryogenesis if applied with a cytokinin-like compound (TDZ + BAP) [140]. SA’s effect on inhibiting cell-to-cell communication and reducing water content in plant cells is an effective way to increase cell viability and dehydration tolerance required for successful cryopreservation [141,142].

##### Abscisic Acid

Abscisic acid (ABA) is linked to the “stress hormone”. Endogenous ABA plays a role in seed maturation (desiccation). Exogenous ABA application is useful for de novo embryo/shoot induction [143,144,145] by the initial induction of auxin accumulation and further cell re-programming (auxin biosynthesis and epigenetic regulation). Exogenous ABA modulates the auxin/cytokinin effect in PTC and leads to an increase in the SE rate in barley [143].

##### Gibberellic Acid

The gibberellic acid (GA) effect on PTC was first described in 1958 [146]. This hormone promotes cell expansion and can be used at low concentrations during shoot proliferation in combination with cytokinin [147]. However, if conditions are optimal, endogenous hormone synthesis should be enough for shoot development. GA biosynthesis inhibitor paclobutrazol (PBZ) showed a positive effect during de novo organ induction and can be used for somatic embryo or shoot induction by pulse application [148].

##### Ethylene

Ethylene is an important regulator of plant development, and its actions were described a relatively long time ago [149,150,151]. Ethylene can be considered a typical stress hormone [152]; therefore, it has a dual role during plant morphogenesis in vitro. Ethylene should get special attention in PTC since plant growth is relatively small in isolated volumes with lower air exchange rates than in natural conditions, a system with a high risk of ethylene accumulation. During the induction step, high ethylene concentration has a positive effect on somatic embryo induction [131] by promoting auxin biosynthesis and the inhibition of transport [153,154]. Moreover, ethylene also influences mineral nutrition through auxin signaling [155,156]. However, in the shoot proliferation stage, ethylene has a number of inhibitory effects: inhibition of shoot growth, hyperhydricity, etc. That is why ethylene accumulation should be reduced by increasing the air exchange rate and using ethylene inhibitors like silver nitrate (AgNO_3_) or cobalt.

### 2.5. Culture Conditions

#### 2.5.1. pH as Growth Regulator

pH is one of the most potent regulators of plant development since it directly regulates transport, including auxin transport. Most of the reactions in plant cells are pH dependent; therefore, pH must be precisely balanced. There are three different types of pH in PTC: cytoplasmic pH (less or more constant, around 7, very buffering); extracellular (medium) pH, and vacuolar pH. According to the vacuolar pH value, one can distinguish three types of vacuoles: alfa type (PSV—protein storage vacuole), associated with stem cells in both shoot and root; beta type, associated with dividing cells; and gamma type (lytic vacuole), associated with fully expanding/apoptotic cells. The transition from beta to alfa type can be associated with the formation of shoot stem cells. Local auxin accumulation may serve as a factor that induces PSV. On the other hand, chloride can promote the induction of a lytic vacuole by rapidly transiting to the vacuole, inducing acidification and subsequent water uptake.

The most manipulable pH is medium pH. Slow artificial medium acidification promoted plant morphogenesis [58]. In addition, all plant species can be divided into acidic, neutral, and alkaline, depending on optimal soil growth conditions. More details on this classification are needed. Barley and sugar beet are typical alkaline plants, and rice is acidic. Alkaline plants induce extremely rapid external medium acidification by high proton efflux from the plants to the medium. This efflux, in turn, may induce cytoplasm alkalinization. Alkalinization can be a main problem related to plants growing in soil with a high pH in nature and is generally recalcitrant for plant morphogenesis in vitro.

In natural conditions, plants must create an acid (local) environment to facilitate nutrient uptake. A recent investigation pointed out the key role of pH homeostasis in the plant morphogenetic process [157]. In plant tissue culture, for the maintenance of a pH balance, in most cases, researchers use a chemical buffer to keep the pH of media in the physiological range. As an example, we cite the addition to the medium of 5–10 mM MES [2-(N-Morpholino)ethanesulfonic acid] buffer but never controlled the pH after cultivation. Another essential factor that drastically affects the pH of culture media is the addition of nitrogen, ammonium (NH_4_^+^), and nitrate (NO_3_^−^). Ammonium absorption by plant cells might occur in a higher pH environment and is more metabolically efficient than NO_3_^−^, requiring relatively less energy for its assimilation. However, this, in turn, is coupled with the H^+^ efflux, which will rapidly acidify the culture medium, and NH_4_^+^ might easily reach toxic levels within plant cells. NO_3_^−^, in turn, might be more absorbed at acidic pH levels in a more controlled way by plant cells, and this nitrate uptake releases OH^–^, again increasing culture medium pH. The carbonic acid/bicarbonate buffer is considered the most critical system for cell pH homeostasis.

#### 2.5.2. Light

Light is a key environmental factor of plant-regulated growth. In PTC, the actual light intensity provided by a fluorescent lamp in the growth chamber is up to 100–120 µmol/m^2^/s. However, natural sunlight intensity may be high at 2000 µmol/m^2^/s. In the shade illuminated by a clear blue sky, midday − 20,000 Lux = 400 µmol/m^2^/s. Moreover, if very little shade occurs in nature (field), significant shade can occur in a standard growth chamber. Light can also induce high reactive oxygen species (ROS) production in the callus, and this ROS can serve as the most simple and powerful tool for shoot induction. The simplest method is to insert plates/tubes during induction for strong light (700–800 µmol or direct sunlight) for 1–2 h and return to regular light (100 µmol) after that.

In many cases, such tricks can increase shoot formation efficiency as a temporary stress factor, which does not require transfer to the new medium. Also, in line with this, light might have the same effects as a morphogen (i.e., direct influence on plant morphogenesis) or as a stress factor for plants growing in vitro. Usually, up to 100 µmol/m^2^/s would have an effect as a morphogen, as photosynthesis in micropropagation conditions is irrelevant. In addition to the amount of light, different light spectrums significantly affect plant tissue culture. Many light-emitting diodes (LEDs) sources with different ranges are currently available and need to be considered as energy spectra [158]. One of the possible mechanisms of different light source effects is an alteration in the endogenous hormone levels [159]. These factors also need to be considered.

### 2.6. Stress Factor and Epigenetic

#### 2.6.1. Stress Factors

Stress factors (but not stress by itself) have a dual role in PTC. In plants, stress-induced agents inhibited cell-to-cell communication [160] and induced branching both in roots and shoots through re-distribution of auxin signaling. This, in turn, lead to an increase in plant regeneration ability [161]. ROS also play a role in stem cell induction and fate regulation (for a review, see [162]). Moreover, the positive effect of hydrogen peroxide on regeneration capacity has been reported as well [163]. On the whole plant level, the application of stress factors leads to hormone re-distribution and, through it, the induction of several epigenetic changes locally, differently in different cell types. In PTC, stress-inducing factors are key for morphogenesis induction through creating the organizer cell niche by the initial accumulation of high auxin contents, the induction of histone hyperacetylation, and auxin biosynthesis induction in specific cell types [164]. In this case, with a few exceptions (i.e., heat shock on microspore embryogenesis), this effect of the stress-induced agents can be achieved only in combination with exogenous 2,4,-D [165] or cytokinin (TDZ) [131]. Stress factors in general play a key role in embryo induction during microspore embryogenesis through changes in epigenetics and, therefore, auxin homeostasis [166]. Paraquat, alloxan, and menadione promote compact cell division and the formation of colonies independent from exogenous auxin [76,165]. Conversely, the inhibition of ROS production with diphenylene iodonium (DPI) or scavenging ROS with N,N′-dimethylthiourea (DMTU) or ascorbic acid prevents cell cycle and shoot induction [165].

However, it is important to note that stress factors should be used very carefully since they inhibit shoot growth. This means that stress factors should be applied only briefly during de novo stem cell induction. Once stem cells and primary shoots are induced, stress factors need to be removed for proper shoot outgrowth. Continued application of stress factors leads to numerous small shoots that do not expand and tend to form calluses because of auxin overaccumulation in young leaves.

#### 2.6.2. Epigenetic Regulators

Epigenetic regulation plays a central role in all plant morphogenic processes, including plant morphogenesis in vitro [167,168,169,170]. Recently, it was shown that plant morphogenesis in vitro has an epigenetic mechanism similar to that of plants [171]. Epigenetic regulation includes dynamic changes in DNA methylation, histone modification, chromatin accessibility, and other processes that determine the transition of somatic cells to embryogenesis [166,172,173]. However, “direct epigenetic regulators” were rarely used in PTC since during morphogenesis, the process of DNA methylation/histone modification is very dynamic and, in many cases, it occurs differently in different types of plant cells during embryo/shoot induction [172]. As an example, we can cite [173], which claimed that mutation in the histone deacetylases (HDAC) gene inhibited plant regeneration. This is very logical since histone hyperacetylation is required only for the formation of an organizer stem cell niche, and thereafter, histone performs deacetylation to form leaf primordia. So, constitutive histone hyperacetylation inhibited shoot growth. In this case, applying certain compounds for all cells in forming morphogenesis will negatively affect the de novo shoot/embryo. Among direct epi regulators, only Trichostatin (TCA) is widely used mainly in pollen embryogenesis since histone hyperacetylation is a key step in organizer cell induction, and there are no other effects since pollen is a single-cell system, [174] and references therein. Some authors have applied 5-Azacytidine, a demethylation agent in the cell cycle, and increased morphogenic responses were observed on a regeneration level [175]; however, the most powerful and suitable epigenetic regulator is phytohormone auxin: application of exogenous auxin/cytokinin on a single cell level significantly changes chromatin organization in plant cells and, therefore, leads to cellular re-programming [176,177]. In plants, auxin gradient formation during de novo organ induction is key for cell fate specification and chromatin remodeling.

### 2.7. Plant Medium Preparation—Focus on Ions Chemistry

Typical plant media contains 12–14 ions, which have their own chemistry and potentially can interact with each other to form precipitates [88]. One of the most problematic ions is iron because it has relatively poor solubility. To improve iron efficiency, the majority of the medium iron is prepared as chelate. However, one should consider that complexes readily dissociate in solutions with a pH level less than 6, and EDTA can easily chelate other ions. Besides Fe, Ca, Mg, Cu, Zn, and Mn can precipitate during cultivation [178]. Altogether, such chemical behavior creates a serious problem with nutrient availability and may lead to poor plant growth and morphogenesis. It is fascinating that in freshly prepared media, the concentrations of free divalent cations like Cu, Ca, etc., were much less than those added during medium preparation because they exist in the form of chelates. However, during plant cultivation, most ions become free and can be absorbed by plant cells. In the natural soil, this problem has been solved by the presence of organic matter from complexes with divalent cations. In a culture medium, adding a trace amount of organic matter can significantly help and ultimately prevent precipitation even after autoclaving. This is especially important for the liquid medium for bioreactors and high-volume suspension culture. We highly recommended adding 100 mg/L of Bacto-tryptone (casein hydrolysate) to the medium during preparation. In this case, divalent cations form complexes with amino acids, which prevent precipitation and keep all ions accessible to plants. Another factor that significantly affects nutrient availability is pH. A more detailed description of this phenomenon can be found in the pH section [179].

### 2.8. Competent Explants

#### 2.8.1. Choosing the Right Explants

Corresponding explants are the key to success in PTC. There are dramatic differences between monocotyledon in dicotyledon plants since in the case of monocotyledon cells, they were very rapidly terminally differentiated after exiting from the meristematic zone (except intercalary meristem, which formed after spike induction). That is why the best initial explant for cereal is scutellular tissue from immature embryos or leaf base from young seedlings. Interestingly, cereal leaves (contrary to dicotyledon leaf) showed an auxin transport direction from the base to the tips, suggesting an absence of auxin synthesis in cereal leaf which is the main inducer of cell division and further morphogenesis.

For the dicotyledon plants, leaf/cotyledon serve as an auxin source, and auxin transport is directed from leaf tips to the stem/hypocotyl and roots. That is why dicotyledon leaf/cotyledon were able to induce cell division and, in some cases, de novo shoots also in natural conditions. Leaf and cotyledon may serve as good/optimal explants for de novo shoot induction. Cotyledon has some advantages compared with leaf since it is a more standard system, while the ability of leaf regeneration is dependent on many factors that need to be considered. Cotyledon can serve as the most suitable explant for many dicotyledon species [134,180,181] because of the high capacity of auxin synthesis through the IPA pathway [182]. However, cotyledon is competent for plant regeneration only in a relatively short “window”: before the induction of shoot apical meristem. At this stage, cotyledon serves as a main auxin source for root growth. However, once shoot induced, the cotyledon is subjected to very rapid differentiation with endoreduplication, chromatin condensation, and limited auxin synthesis capacity. From this moment, cotyledon switches to photosynthesis as its main task and, therefore, is not competent for plant regeneration anymore.

The hypocotyl is part of the stem of young seedlings below the cotyledons and roots. In seedlings, the hypocotyl serves as a potential source of adventitious roots and shoots. The system of plant regeneration from the hypocotyl explants was established a long time ago [180,181]. The main features of the hypocotyl are local auxin metabolism [182] and a “sink “for the cotyledon-derived auxin. Currently, this regeneration system is widely used in plant tissue culture [183,184]. Recent investigations of gene expressions in cotton hypocotyl regeneration systems suggest that auxin synthesis and polar transport are essential for de novo embryo/shoot induction [185]. There are two possible regeneration protocols: new shoot induction by the application of exogenous cytokinin (BAP/kin) or on hormone-free medium by endogenous hormones [186]. In the latter case, hypocotyl-derived auxin first induces adventitious root, and root-derived cytokinin is able to induce de novo auxin synthesis, accumulation, and cell re-programming in the apical part of the hypocotyl. In the case of direct cytokinin application, the shoot originates directly from the competent cell in the epidermis-attached tissue.

In contrary to dicotyledonous plants, monocotyledons have more “compact “sites of auxin synthesis and cell cycles correspondingly. In most cereals, this site is restricted to just a few millimeters from the shoot–root junction with an auxin transport direction from the shoot–root junction towards the leaf tips. This also explains why cereals have a very restricted morphogenic capacity, and ideal explants for plant regeneration are immature organs at a very specific stage (expanding but not differentiated), as, for instance, scutella from 12- to 14-day-old wheat seedlings are the most suitable. Recent investigation shows that scutella have a very high auxin metabolism [187,188].

Plant regeneration from callus requires special knowledge and some tricks. Callus is an unorganized cell mass in which the majority of cells are equal and require exogenous hormone (auxin/cytokinin) application or endogenous hormone production [115,116]. In many cases, callus originates from certain cell types (mesophyll cells in some dicotyledonous species, for example) that have high ploidy levels and cannot regenerate diploid plants. In addition, during callus culture, a lot of somaclonal variations and other abnormalities can be induced, which, in turn, lead to plants with characteristics diverging from their mother plant [189,190].

Altogether, this makes a callus a difficult object for true-to-type plant regeneration in the majority of species (except model objects like tobacco and Solanum nigrum, for example). The main pre-request of de novo shoot formation is the induction of organizer cells with specific features (auxin source plus relative dormancy because of histone hyperacetylation). Only this type of cell is responsible for creating a de novo auxin gradient and developmental gradients. In this case, applying pulse stress factors in combination with exogenous auxin/cytokinin may trigger the creation of an “organizer cell” from the callus. However, this step must be performed very carefully, and treatments must be designed for each type of callus/explant individually.

In plant tissue culture, callus culture has another practical approach: the production of secondary metabolites for the pharmacological industry and the production of biologically active compounds [191,192,193]. The most important feature of this method is the possibility of regulating the level of bioactive compound synthesis relatively easily by changing hormonal signaling [194]. The modern methods of gene editing by CRISP-GAS open up the possibility of producing bioactive compounds by callus culture since they allow us to avoid several steps, such as plant regeneration, and directly use transgenic cells as a bioactive compound source at the industrial level.

#### 2.8.2. Rooting of New Regenerated Plants

Rooting of de novo forming shoots is an essential step in plant tissue culture for completing the micropropagation step and performing successful plant adaptation in soil conditions [195]. In some species, this step is limiting of the whole procedure. In order to avoid these difficulties and perform successful rooting, one needs to understand the physiological mechanism of this morphological response. From a physiological point of view, through the root is a natural means of canalization of the shoot-derived auxin.

There are three main pre-requests of successful rooting: high auxin production in certain xylem-adjusted cells in the stem. The presence of vessels (sieve elements) is key for plant rooting since only this system allows the creation of enough powerful channels for auxin transport and the open ability to canalize through roots. Several protocols for de novo root induction have been used [196], which include pulse treatments with relatively high auxin contents (IBA) and darkness. In this case, two possible problems in natural rooting have been solved: tissue (stem) with vessel elements gets higher auxin content and was able to induce temporary auxin maximum to activate cell division in a xylem-pole cell with de novo root induction.

#### 2.8.3. Hyprehydricity

Hyperhydricity is a common physiological disorder in de novo-formed shoots [197,198]. The main physiological reason for such a disorder is an imbalance between new cell formation and growth. The molecular basis of this phenomenon is multiplying auxin functions. Auxin is responsible for cell division and cell expansion. In plants, this role is separated by different auxin synthesis pathways located in different cell types. Once the shoot grew, the multiply auxin biosynthesis pathway was activated. In this case, high auxin production in the mesophyll cell may lead to very rapid cell expansion through vacuolar growth. This leads to excess water uptake and cell deformation. In turn, excess auxin in the mesophyll cell cannot canalize through vascular tissue and inhibit rooting. There are several basic tricks to prevent excess water uptake: reduce auxin production or increase the rate of auxin efflux. One of the possible methods to reduce hyperhydricity is to reduce the ethylene level with silver nitrate [199] or by increasing the air exchange rate. The next trick is nutritional balance: lower nitrogen and chloride. At this stage, chloride is rapidly absorbed by the cell, increasing vacuolar acidity and water uptake. The final method is a reduction in exogenous cytokinin which, in turn, will reduce the auxin biosynthesis rate.

## 3. Conclusions

The current review systematically analyzed major aspects of plant growth regulation in tissue culture in vitro, including the nutritional aspects of inorganic and organic additives as possible interactors with endogenous hormone metabolism. Among different phytohormones, only endogenous auxin can be transported polarly and is able to induce polarity. Therefore, auxin has been postulated as a major factor regulating plant morphogenesis in vitro, while cytokinin may serve as an auxin biosynthesis inducer [200]. The effect of endogenous auxin is dependent on the pathway of auxin biosynthesis and the cell type and, therefore, the auxin canalization pathway. Auxin canalization through vascular tissue led to root formation, while active auxin transport mediated by auxin transport carriers led to shooting formation if the tissue kept a balance between auxin production and canalization. We also pointed out the pivotal role of auxin accumulation as a starting point for de novo shoot organogenesis and the role of stress-inhibited auxin efflux in this process. Among phytohormones, auxin is a key factor that regulates plant morphogenesis in vitro, regulating both shoots (through auxin canalization in a xylem-free system) and roots (through xylem-mediated auxin canalization). IPA-mediated auxin biosynthesis is essential for the induction of organizer cells. Stress factors play a key role in this induction’s initial stage but inhibit shoot proliferation.

Moreover, we pointed out that the plant tissue culture process is regulated epigenetically and is dependent on the epigenetic status of plant tissue. Based upon this knowledge, a simple flowchart (Figure 2) for plant micropropagation involves breaking down the process into sequential steps, and when establishing a PTC protocol for a new species, we suggest a strategic approach that includes understanding the natural growth conditions of the plant (such as soil pH, ion composition, and also seed composition), assessing the epigenetic status of the starting material, and preparing a culture medium that mimics the natural conditions as closely as possible. Observing the initial stages of PTC is also crucial, particularly for signs like the type of callus formation, polyploidization, and the nature of cell division.

## Figures and Tables

**Figure 1 plants-13-00327-f001:**
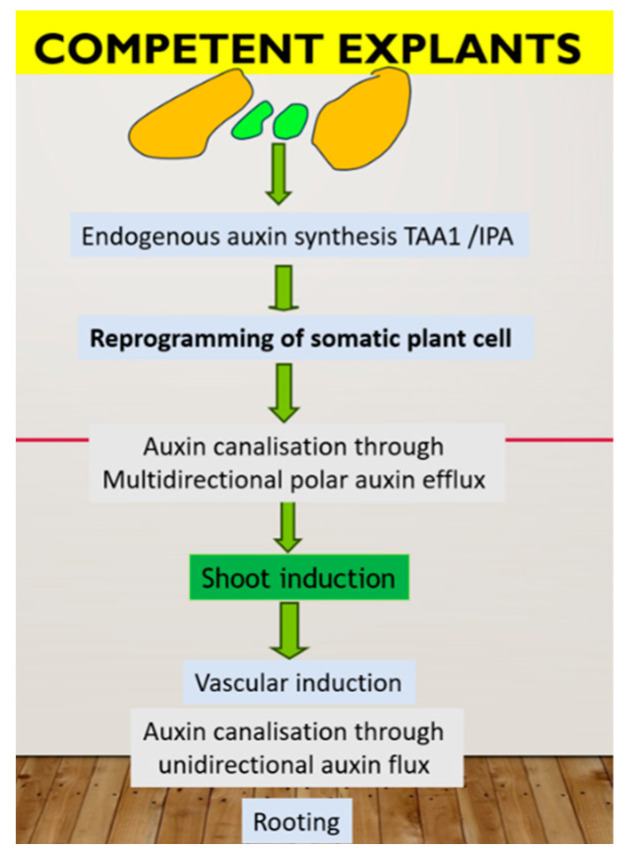
Hypothetic mechanism of plant regeneration in tissue culture in vitro. In green are representative of organizer cell, controlling in vitro response through auxin biosynthesis.

**Figure 2 plants-13-00327-f002:**
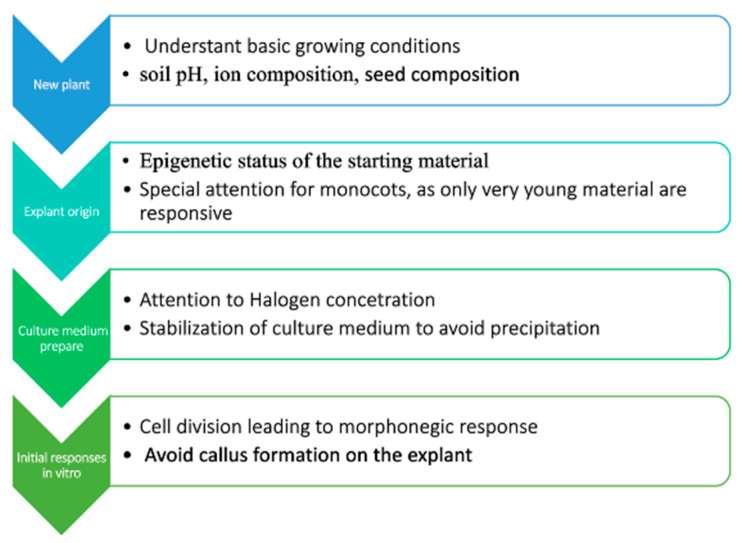
Proposed flowchart for plant micropropagation based upon the characteristics and aspects presented.

## Data Availability

Not applicable.

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
