# Peer review of "Plant Growth Regulation in Cell and Tissue Culture In Vitro"

_plants, 2024, doi:10.3390/plants13020327_

Round 1

Reviewer 1 Report

Comments and Suggestions for Authors

The review is well organized and didactic. It describes the role of nutrients, organic compounds, and hormones in regulating plant morphogenesis in tissue culture. It also provides tips for further optimization of tissue culture systems.

I am suggesting some modifications that could improve the quality of the document before publication.

Some general comments:

- Many sentences require the addition of references to support the information presented - see the point-by-point review below.

- Introduction: I missed a paragraph that briefly describes the route (strategy) that is used when we are working on the in vitro culture of a species that has already been well studied in vitro and a new species. In other words, what is the decision-making process when we start a new in vitro culture - especially for species that have not been well studied and for recalcitrant ones. I suggest adding a paragraph on this and perhaps a diagram containing the main points that should be considered in these cases.

- I really miss a flowchart demonstrating all these aspects raised. I think this would enhance this review even more.

- Conclusion: I strongly suggest adding a concluding paragraph that addresses the main contributions of this review for future investigations using tissue culture and what are the future perspectives for the use of this important and practically indispensable tool.

 - Please see my detailed comments/suggestions below:

Line 4: Add affiliation, address, etc.
Line 14: "on several factors such as  xxx". I suggest listing some main ones here.
Lines 14-16: Consider reviewing it to: In this review, the most important aspects are described one by one, with some practical recommendations based on basic research in plant physiology...
Line 23: Consider deleting the word "across"
Line 43: industries
Line 56: Add a reference to support this sentence.
Line 62: Consider adding some references instead of "(Several authors)"
Line 68: In the root, the stem cell niche is formed as a result of..  How about the pericycle?
Line 69: through the polar xylem mediated (as an initial stage)
Lines 68-73: Add references to support these 2 sentences.
Lines 80-91: It is necessary to add references to this topic.
Line 94: Describe abbreviations at first mention, e.g. "SE", 2,4-D (Line 96), SA, Fe, Cu (Line 107), N-K-Cl-Ca-S-Mg-P-Fe-Mn-144 B-Zn-Mo-I-Cu-Co with N:P (Lines 144-145), NH4H2PO4 (Line 246)
Line 94: Consider rephrasing the sentence, as in some cases this variation is interesting.
Line 96: Add a reference to support this sentence.
Line 97: Consider reviewing it to: These synthetic auxins, in addition to their auxinic effect, bind to F-box proteins as transport inhibitor response 1 (TIR1) [7] and can induce ...
Lines 101-102: the "organizer cell", which in the future will divide at the periphery to form a globular structure with epidermis around it (Consider adding a reference to support it).
Line 102:  such a
Line 102-103: Consider reviewing it to: competent cells also start dividing and form callus due to exogenous auxin.
Line 107:  It is necessary to add references to support this sentence.
Lines 112-119: Same comment as above!
Line 123: it is especially
Line 124: Reference needed.
Line 127: The sentence appears to be incomplete; please check it.
Line 130: Reference needed.
Line 134: It is not necessary to add the authors' surnames, just the numerical reference.
Line 140: How about the phase that the culture is in in vitro? i.e. it can be related to the phase (initiation, regeneration, multiplication, rooting, etc.).
Line 148:  "halogenide" or "halide"... please check it.
Lines 150-152: Consider reviewing it to: Nowadays, several authors, who also work on the optimization of the composition of the culture media, have also suggested that....
Line 155: Reference needed.
Line 179: regeneration. Sucrose is ....
Line 184, 188 and 202:  Reference needed.
Line 202: Consider adding a sentence on hyperhydricity X nitrogen
Line 210: A recent review of xxx [29]
Line 220: a negative effect on xxxxx
Line 221: Reference needed.
Lines 226-227: There are many reports how phosphate nutrition regulated processes in PTC (OK, but add some references and describe the abbreviation)
Line 231: Calcium plays an important role in plant tissue culture - please explore this further!
Line 233: Reference needed.
Lines 234 and 237: CaCl2, Ca(NO3)2 - describe, please..
Line 235: In the majority of the cases, authors used ..... Ok, so it is necessary to add a couple of references then!
Line 238: "as a minor nutrient." I got what you're saying, but it's not clear.
Line 243: MgSO4 - OK, but add a reference to support this sentence and describe the abbreviation)
Line 244: Either deficiency or excess of Mg has... (Reference needed).
Line 247: Please explore this last sentence further; there seems to be a missing ending.
Line 248: How about the source of iron? Consider adding a comment on that!!
Line 252: is 70-100 μM - Reference needed!
Line 259: The primary function of boron is cell-to-cell communication - Reference needed!
Line 260: Reference needed!
Line 265: "this is not very easy," - Check the grammar of the last sentence
Lines 271 and 302: "Very complicated" - please explain it further!
Line 278: 0.2 to 1 μM - Reference needed!
Line 283, 291 and 301: Same comment as above.
Line 288: SOD??
Line 295: This information must be referenced.
Line 305: "„" ??
Line 309: „function„ ??
Line 310: Reference needed!
Line 312: Kao medium???
Line 318: Reference needed!
Line 323: „buffer„ ??
Line 333: ASC? Describe the abbreviation the first time it is mentioned.
Line 336: (ref) ???
Line 340: DHA?
Lines 336-346: No references added for all this information! Please add references to support this information.
Line 373: Delete "for review".
Line 373: "Linsmaer and Skoog (1965) demonstrated" - add the numerical reference...

Line 378: "[78] Digby, J., & Skoog, F. (1966)" Please review it!
Line 385: "1-2 mg/l of B6 is a standard concentration in the medium." Reference needed!
Line 386: " vitamin stress" may positively affect PTC - please explore it further!
Line 395: "for all processes in PTC" - please review it and add some examples and references!
Line 396: Reference needed!
Lines 403-404: ANA/AIA - please describe them!
Line 414: IPA?
Line 442: Reference needed!
Lines 446-447: BAP, kin, 2-IP, Zea, Zea-R  - please describe them!
Lines 456, 462 and 465: Reference needed!
Line 466: Please check this "„" symbol through the manuscript - what does it mean?
Line 468: It is not necessary to add the authors' surnames, just the numerical reference.
Line 469: Delete "for review".
Line 480: I strongly suggest adding a sentence mentioning recent investigations on the use of salicylic acid in culture media and/or solutions during shoot tip cryopreservation procedures. Some references for your consideration (https://doi.org/10.21273/HORTSCI13958-19 ; https://doi.org/10.29312/remexca.v10i7.1627 ; https://doi.org/10.1023/A:1014416817303)
Line 482: Reference needed!
Line 494: Consider just adding the numerical reference instead (for review¨Abdalla, N., et al., 2021)
Line 507 and 519: AgNO3, PSV - describe please.
Line 522: "The easiest manipulable is medium pH." Please explore it further!
Line 526: "beet. And" - Double check the grammar/punctuation.
Line 536: "MES" - describe please.
Line 536, 540 and 544: Reference needed!
Line 544: Consider adding a comment on duration of autoclave time X pH,  container size x pH, culture age x medium pH.
Lines 546-554: Do you think it is not necessary to add references here? There is a lot of information in this section. Line 570: "ROS" - describe please.
Line 601: "3" - the number 3 is lost in the section... Please delete it.
Line 602:  Temman, H. et al. 2023 [140]
Line 604: Double check the punctuation of the sentence.
Line 629, 649 and 651: Reference needed!
Line 643: monocotyledon and dicotyledon
Line 695: Reference needed!
Line 744: How about the nitrogen level in the culture medium?  
Line 744: I really miss a flowchart demonstrating all these aspects raised. I think this would enhance this review even more.
Line 756: I strongly suggest adding a concluding paragraph that addresses the main contributions of this review for future investigations using tissue culture and what are the future perspectives for the use of this important and practically indispensable tool.

Author Response

Dear reviewer,
Thank you very much for your very detailed and constructive comments.
We addressed all of them by adding corresponding citations.
We also added a flowchart as required and extended concluding remarks.
Again, many thanks for your constructive comments.
With my best regards!

Reviewer 2 Report

Comments and Suggestions for Authors

The authors of this manuscript reviewed the role of plant growth regulators in cell and tissue culture.

The manuscript was well-designed and written and summarizes the published work about plant tissue culture. Although the manuscript has basic information that is known to the researchers in the PTC field, it adds very useful practical information and could help anyone who works in the field.

However, there are very few points that might improve the manuscript. Please find the attached pdf file.

Comments on the Quality of English Language

The manuscript was well-written; however, there are very few points that might improve the readability of the manuscript.

Please find the attached pdf file.

Author Response

Dear reviewer,
Thank you very much for your very detailed and constructive comments.
We addressed all of them in a new version.
We also added a flowchart and extended concluding remarks.
Again, many thanks for your constructive comments.
With my best regards!

Round 2

Reviewer 1 Report

Comments and Suggestions for Authors

The authors did a great job reviewing the document. I support its publication in the last version.